# Separating Chickens’ Heads and Legs in Thermal Images via Object Detection and Machine Learning Models to Predict Avian Influenza and Newcastle Disease

**DOI:** 10.3390/ani15081114

**Published:** 2025-04-11

**Authors:** Alireza Ansarimovahed, Ahmad Banakar, Guoming Li, Seyed Mohamad Javidan

**Affiliations:** 1Biosystems Engineering Department, Tarbiat Modares University, Tehran P.O. Box 14115-111, Iran; a.ansarimovahed@modares.ac.ir (A.A.); mohamad.javidan@modares.ac.ir (S.M.J.); 2Department of Poultry Science, University of Georgia, Athens, GA 30602, USA; gmli@uga.edu

**Keywords:** thermography, feature extraction, poultry body temperature, relief feature selection, YOLO algorithm, support vector machine (SVM)

## Abstract

The early detection of poultry diseases is crucial for preventing outbreaks and ensuring animal health. This study utilized the thermal imaging of chickens’ heads and legs to identify avian influenza and Newcastle disease. A deep learning model (YOLOv8) was applied to separate the head and leg regions in the obtained thermal images, followed by a classification model (support vector machine) to predict infections. This study analyzed three types of images: original, background-removed, and those with extracted head and leg areas. The results showed that focusing on these regions improved the diagnosis accuracy, with over 90% accuracy being reached within just 8 h of infection. The method was more effective in detecting Newcastle disease than avian influenza. The findings suggest that this non-invasive approach can help poultry farmers and veterinarians detect diseases early, improving disease control and reducing economic losses.

## 1. Introduction

Among the various poultry diseases, avian influenza and Newcastle disease are considered among the most dangerous and costly diseases [1]. The influenza and Newcastle disease, in poultry, have caused significant mortality and substantial economic losses for the poultry industry. Newcastle disease is a contagious disease among chickens and is prevalent in both domestic and wild species. This disease can also be transmitted to humans [2]. The disease was first identified in Newcastle, England, in 1926 and has been known by this name since then [3]. The symptoms include respiratory and neurological signs, hemorrhagic lesions in the digestive system, and mild or minor respiratory infections. The causative agent of Newcastle disease is a single-stranded RNA virus that is a member of the Paramyxovirus family. Avian influenza is another contagious respiratory viral disease in poultry that has a rapid spread rate [4]. The influenza viruses belong to the Orthomyxoviridae family and the influenza virus genus. These viruses are divided into five genera based on the antigenic characteristics of their internal proteins and matrix proteins, among which type A influenza viruses are more significant in terms of their pathogenicity [5].

In the poultry industry, rapid and accurate disease detection, in addition to vaccination and nutrition programs, is highly important to enabling quick and appropriate responses to disease outbreaks and reduce their resulting damages. The implementation of vaccination programs is one of the most important health measures in poultry farming, and the nutrition of animals is an environmental factor that plays a crucial role in their immune response during vaccination. It can improve or reduce the efficiency of vaccines, or even eliminate their effectiveness [6]. Nutritional factors that affect the immune response to vaccination have been separately examined in numerous articles; however, unfortunately, there are few reports that comprehensively present these factors in a single article.

According to previous studies, the use of technologies related to sound and image processing can be beneficial in identifying poultry diseases [7]. Zhuang et al. (2018) were able to identify healthy and sick birds infected with H5N2 influenza virus using image processing [8]. The background was removed from raw images, and the body position features (arching, standing, pecking) were then extracted. The machine learning classifier that they used, a support vector machine model, was able to classify healthy and sick birds with 99.46% accuracy. Minna et al. (2018) [9] could differentiate between healthy chickens and sick ones (infected with jaundice) through color changes in tissue at specific points, including the chicken’s eye and comb, when it was affected by disease. The accuracy results obtained by the support vector machine classifier were 92.5%. Wang et al. (2019) [10] studied 10,000 Ross chickens aged 25–35 days and were able to diagnose digestive disease through features such as the shape, color, water consumption, and feed consumption of the chickens. The model that they used was a convolutional neural network, which achieved 93.3% prediction accuracy. Okinda et al. (2019) [11] predicted Newcastle disease in broiler chickens using image processing with the extraction of the features of variance, kurtosis, convexity, and complexity. They showed that healthy chickens move significantly faster, between 0.15 m/s and 0.17 m/s, while sick chickens move slower at under 0.11 m/s. In their study, a support vector machine classifier with the radial basis function kernel achieved 97% accuracy with a significance level of 0.05. These methods show potential in using image processing and machine learning for poultry disease predictions.

Deep learning algorithms have been introduced for poultry disease detection. Degu et al. (2023) [12] used two main algorithms, the You Only Look Once (YOLO) V3 object detection algorithm and the ResNet50 image classification model, to detect three diseases in chickens: coccidiosis, salmonella, and Newcastle disease. YOLO-v3 was used to detect regions of interest from fecal images, while ResNet50 was used for classifying segmented images of coccidiosis, salmonella, and Newcastle disease. The models were trained on 1500 collected chicken fecal images. The YOLO-v3 object detection model achieved an average accuracy of 87.48%, while the ResNet50 image model showed a classification accuracy of 98.7%.

Li et al. (2023) [13] developed an on-site feces image classifier system for assessing the health status of chickens. The modified MobileNetV2 model, with additional layers of artificial neural networks, was selected after a comparative evaluation of six image classification models. The system achieved over 90% accuracy in identifying healthy or diseased birds infected with coccidiosis, salmonella, and Newcastle disease, and the whole operational procedure took less than one second. The abovementioned studies were mainly based on normal images or RGB (red-green-blue) images.

Thermal imaging is another non-invasive method that can accurately display temperature variations on the surface of animals’ bodies [14]. Since many diseases, including avian influenza and Newcastle disease, can cause changes in the body temperature of birds, this technology can serve as an effective solution for early disease detection [15]. Alongside thermal imaging, recent advancements in artificial intelligence and machine learning have enabled the automated processing of these images and the identification of patterns associated with diseases [16]. The combination of thermal imaging and machine/deep learning has the potential to allow for the development of intelligent disease detection systems that are capable of quickly and accurately identifying diseases in their early stages. Wilcox et al. (2009) conducted thermal imaging of the footpads of 150 randomly selected chickens using a thermal camera and were able to clinically screen and identify birds with footpad dermatitis [14]. Edgar et al. (2013) [17] investigated changes in the surface body temperature of chickens touched by humans. They focused on three areas, the eyes, comb, and head, and captured infrared thermal images of these regions. Their results showed that, after human touch, the temperatures of the eyes and comb decreased, while the head temperature increased for up to 20 min. Moreover, no correlation was found between physical activities like walking around the cage and body temperature. Sadeghi et al. (2023) [18] successfully identified Newcastle disease and avian influenza in poultry using whole-body thermography and image processing. Four groups of 20 Ross 308 broiler chickens were used in their study. After noise and background removal, 23 statistical features were extracted from each image, and the most relevant features were selected using the distance development evaluation method. Support vector machines and artificial neural networks were employed as classifiers. The best accuracy was achieved with the former model, which detected Newcastle disease and avian influenza 24 h after virus inoculation with accuracies of 81.48% and 97.22%, respectively.

While previous studies have offered the potential for disease detection using thermal imagery and machine/deep learning methods, these methods were applied to entire images. Entire images can include many unnecessary pixel areas (e.g., food bowls, water bowls, and feathers on the floor) that can distract the model’s attention and keep it from looking at the most important features for prediction. The whole body temperature of chickens can be affected by their feather coating, as thermal image cameras capture mainly surface temperature.

Several studies have emphasized the importance of specific areas of the body in using thermal imaging to diagnose disease. McCafferty (2012) [19] showed that thermal imaging, as a non-invasive, non-contact method, can effectively monitor chicken diseases by focusing on heat exchange areas such as the head and legs, which were observed to be significantly affected by diseases. Likewise, Cangar et al. (2008) [20] conducted a comprehensive study of the temperature distribution in different areas of the bodies of chickens and found that areas without feathers, including the head, cheeks, and inner thighs, exhibited the highest rate of heat exchange, while covered areas such as the wings and chest maintained lower temperatures. This variance in temperature distribution indicates that the head and feet are among the most sensitive areas in the diagnosis of infectious diseases such as avian influenza and Newcastle disease. Moreover, since the feather coating can alter surface temperature readings, focusing on the head and legs increases the detection accuracy and reduces the noise in thermal imaging data. The findings confirm that targeting these specific areas in thermal image processing not only improves the accuracy of the diagnostic model but also facilitates the diagnosis of previous diseases. The structure of this paper is as follows: Section 2 describes the materials and methods used in this study, including the sample preparation, imaging process, and data analysis. In Section 3, the results are presented and the performance of the proposed method in the classification of diseases is discussed. Finally, Section 4 summarizes the results and proposals for future research.

## 2. Materials and Methods

In the present study, thermal imaging and machine learning techniques were utilized to detect Newcastle disease and avian influenza in broiler chickens. The major implementation steps included: (1) preparing samples and inoculating diseases for birds; (2) acquiring thermal images; (3) separating chicken body (foreground) from the background via image processing; (4) applying object detection models to isolate the head and leg of each chicken; (5) extracting critical features; (6) using sparse matrices to reduce noisy data; (7) selecting essential features; and (8) classifying diseases. The analyses were conducted at different stages of disease transmission using three sets of data: original images, background-removed images, and images with chickens’ heads and legs only. Figure 1 provides a comprehensive depiction of the steps undertaken in this study.

### 2.1. Preparation of Samples and Disease Inoculation

A total of 80 Ross 308 broiler chickens were used in this study. The day-old chickens were raised in the poultry house of the Faculty of Agriculture, Tarbiat Modares University. At day 14, the chickens were transferred to Kashkak village, Aligudarz County, Lorestan Province, Iran for disease challenge experiments.

The experimental environment consisted of electric heaters, a hygrometer, a thermometer, an air ventilation blower, and separate experimental pens. The pens were equipped with drinkers and feeders. The 80 chickens were divided into four groups, and each group consisted of 20 birds and was examined separately. For each group, 20 experimental pens (each was measured at 0.8 m wide × 1.2 m long) were constructed. These pens, with one bird per pen, were separated by metal wire to prevent contact among the chickens. The pen floor was covered with sawdust to provide comfortable bedding for the chickens.

After the chickens were placed in their respective cages, they were allowed three days to adapt to the environment to reduce their stress. Then, the virus was injected into the eye (through a dropper) and imaging was done every 8 h until the manifest symptoms of the disease appeared. During the experiment, the chickens were fed with a commercial standard ration for broiler poultry. The ambient temperature was maintained at 25 ± 1 °C and the relative humidity was between 50 and 60%, and these conditions were monitored daily to check the health of the poultry.

In order to conduct tests and inoculate the virus, the room was first disinfected using steering bricks and germicide, and it was disinfected again after completion of the experiment for each group. Groups 1 and 2 were designated as the control and influenza virus-infected samples, respectively, while Groups 3 and 4 were designated as the control and Newcastle virus-infected samples, respectively. Virus inoculation was performed via the ocular route using droppers, as shown in Figure 2. The volume of the virus introduced into each chicken’s eye was 0.1 cc. To confirm the presence of the virus in the chickens, the RT-PCR molecular test was performed in collaboration with the Razi Vaccine and Serum Research Institute located in the Hessarak district in Karaj, Iran.

### 2.2. Imaging

Temperature and humidity were measured and displayed using a wall-mounted Sigma peripheral sensor (Sigma, Aizu, Japan), ensuring real-time monitoring and control of peripheral parameters to meet the needs of breeding chickens. The Sigma sensor that was used measures temperature-openings in the range of −50–70 °C with an accuracy of ±1 °C and humidity in the range of 10–99% RH with an accuracy of ±3% for 50%−80% RH and ±5% for other ranges.

A FLIR A65 thermal camera (FLIR, Wilsonville, OR, USA; Figure 3) with a resolution of 512 × 640 pixels, a working wavelength range of 7.5 to 13 μm, and a working temperature range from −25 to 135 ° C was used to collect data. The camera had a thermal sensitivity (NETD) of <0.05 ° C at +30 ° C and a temperature accuracy of ±5° C or ±5% of the reading. Initial stretching and processing of thermal images was carried out using FLIR Tools software (ver. 5.13.18031.2002).

To ensure accurate thermal readings, calibration was carried out by measuring the temperature of a reference object with a known temperature and adjusting the other readings accordingly. The distance from the camera to the bird was fixed at 50 cm and, according to, the thermal camera emission coefficient was set to 0.95 (Figure 3).

Thermal imaging was carried out every 8 h based on the findings of previous studies and considerations related to the physiology of the virus. According to Alexander et al. (2012) [21], physiological changes and measurable temperatures in infected birds begin from 6 to 10 h after infection and intensify over time. In contrast, Sadeghi et al. (2023) [18] were able to achieve infected by Newcastle and influenza diseases after 24 h with a diagnostic accuracy of 81.48% and 97.22%. Reducing the interval to 8 h with the aim of recording the progression of the disease more accurately also enabled early diagnosis of the disease. This time frame was chosen as a practical balance, as shorter intervals (e.g., 4 h) can cause excessive stress due to frequent manipulation, while longer intervals (e.g., 12 h) may miss important early changes. In addition, the 8 h interval guaranteed full coverage of day and night cycles (8, 16, 24, 32, 40, 48, and 64 h after infection), allowing the potential effects of daily rhythms on body temperature to be considered.

The onset of visible disease symptoms was determined through direct observation of external symptoms such as changes in behavior, feeding patterns, and physical appearance. In addition, to confirm the presence of the disease, tissue samples were collected and sent to the laboratory after death, and three samples were analyzed in each group.

### 2.3. Image Processing for Chicken Body-Background Separation

FLIR Thermal Studio is an advanced software developed by FLIR Systems (Wilsonville, OR, USA) for analyzing and processing thermal images. This software enables users to upload thermal images captured by FLIR thermal cameras, perform detailed analysis, and generate reports [19]. In this study, a histogram-based thresholding technique was employed in FLIR Thermal Studio to analyze the frequency distribution of pixel intensities and segment the chicken body from the background in obtained images [20]. The histogram typically exhibited a bimodal distribution, where one peak corresponded to the colder background and the other to the warmer chicken body. To achieve optimal segmentation, Otsu’s thresholding method was applied to determine the best temperature threshold, ensuring accurate separation between the chicken and its surroundings.

This technique was able to distinguish between changes in the chicken's body temperature and the background, allowing for accurate extraction of the chicken's body image. Additionally, it reduced thermal noise and improved the clarity of the segmented regions, making further analysis more reliable [21].

### 2.4. Object Detection to Separate the Head and Leg of a Chicken

In this study, the YOLO-v8 algorithm [22,23], one of the popular deep learning algorithms, was used to divide the heads and legs of chickens. Unlike pre-trained models, the YOLO-v8 model was trained from scratch using a dataset of 2240 thermal images, allowing it to specifically learn thermal imaging features to diagnose chicken disease. The dataset was divided into 80%, for training (1792 images), and 20%, for testing (448 images), to ensure a balanced assessment.

Head areas (titled “H”) and legs (titled “L”) were labeled using Image Labeler (MATLAb 2021rb) to train the model. Thermal images from all time points (8–64 h after infection) were included for both chickens infected with avian influenza and those infected with Newcastle disease. Training was carried out without the use of pre-trained weights, and all default configurations of the Ultralytics community framework were applied.

The training process was configured with the following parameters: stochastic gradient descent with momentum (SGDM) optimizer, initial learning rate of 0.001, mini-batch size of 16, and 100 epochs. The trained model was tested on multiple datasets, each containing 2240 images (280 images per group), to assess its generalization performance under different conditions. These datasets included full-body thermal imaging (with both healthy and sick groups), thermal images with background, thermal images without background, and images of the separated head and legs.

For each dataset, a confusion matrix and a detailed performance table were created to analyze the classification results and evaluate the model’s ability to accurately detect and divide head and leg areas. Based on the test performance, the model achieved more than 99% accuracy in identifying head and leg regions.

After development, the model was used to extract the head and leg areas from thermal imagery, providing the exact coordinates of the bounding boxes to support the disease classification process. Figure 4 illustrates examples of detection from different viewing angles.

In the current approach, object detection (YOLO-v8) and classification are implemented as two separate models. In this framework, YOLO-v8 first identifies and isolates the head and leg areas, and then the extracted thermal features are classified in a separate stage. This modular design provides greater flexibility in feature extraction and allows independent optimization of each step.

### 2.5. Feature Extraction

After isolating the possible disease areas on the chicken’s head and legs, for accurate disease diagnosis, image processing techniques and deep learning algorithms were used to extract significant features for accurate disease detection. These techniques included extracting color and texture features from the images. Color features such as the mean, maximum, median, and standard deviation were calculated in different color spaces (RGB, LAB, HSV, and YCbCr). Texture features were extracted from the gray-level co-occurrence matrix (GLCM), including the contrast, correlation, energy, and entropy. A total of 23 features were defined in this study and extracted from the images, as stated in Table 1. 

### 2.6. Removing Noisy Data with a Sparse Matrix

As shown in Figure 4, after separating the head and leg regions from the thermal images via the YOLO-v8 algorithm, black pixels were padded on most of the regions. Black pixels were constantly represented as zero values, which provide less information for disease classification compared to valid pixel values. Meanwhile, they added complexity and increased the computational time of the detection process. Therefore, these data needed to be removed in a way that did not hinder disease differentiation.

A sparse matrix was used during the feature extraction stage to eliminate zero values in the image. A sparse matrix is a matrix with a large number of zero elements. Such matrices are commonly used in various scientific and engineering fields, particularly in large-scale simulations, network analysis, solving partial differential equations, and machine learning algorithms. Special methods have been employed for storing and operating on sparse matrices to optimize memory and computation time [24,25,26]. Previous studies have used sparse matrices in various image processing applications such as compression, pattern recognition, image filtering, feature extraction, and pattern detection [27]. In this study, it was used to remove pixel values outside the head and leg regions. The sparse pattern S(A) [28] of a non-zero matrix (A) is defined in Equation (1):(1)$$S\left(A\right)=\left\{\left(ij\right)\right|a_{ij}\neq 0\;1\leq ij\leq n\},$$
where $a_{ij}$ represents specific elements in a non-zero matrix A.

One of the methods for identifying a sparse matrix is determined by the following inequality. Matrix (A) is sparse if the following inequality holds:(2)$$3\times n\left(I\left(A\right)\right)+3n\left(J\left(A\right)\right)>3\times \left(n+1\right),$$
where n(I(A)) and n(J(A)) represent the number of rows and columns of matrix A, respectively. In this research, the zero elements of the sparse matrix had the same values as W(A) = 0, which is further explained in the section on the Relief algorithm.

To ensure that the model generalizes well and to mitigate the risk of overfitting, we employed five-fold cross-validation during the evaluation phase. This method helped in verifying that the model performs consistently across different subsets of data, making the results statistically reliable and reducing the dependency on any specific partitioning of the dataset.

We divided the data into five equal subsets, and the model was trained on four subsets and tested on the remaining subset. We repeated this process five times. This ensured that every sample was used for both training and validation, enhancing the stability and robustness of the model’s performance.

### 2.7. Feature Selection

Feature selection plays an important role in improving classification outcomes and reducing computational complexity. Using a large number of features can cause classification confusion [29]. The Relief algorithm was used to select important features. The algorithm ranks the features by quality and helps to eliminate redundant and noisy features [30].

As a filter preprocessing method, Relief evaluates data based on feature differences in the range [0, 1] and estimates the importance of each feature [31]. The algorithm obtained the weight vector associated with each attribute by repeating the learning process several times, and the attributes whose weight exceeded the defined threshold were selected. Pseudocode of the Relief algorithm is presented in Algorithm 1. The iteration loop of the algorithm consisted of m to the random training sample (Ri), where Ri is defined as the target and cable-based sample. W represents the difference between the feature values of the target and neighboring samples. Therefore, in each loop, the distance between the target sample and the other samples was determined. The Relief algorithm selected two of the nearest neighboring samples: one with the same class called nearest hit (collision-H), and the other with the opposite class called error(M) [32].
**Algorithm 1.** Pseudocode of the Relief algorithm [33].Input: for each training instance a vector of attribute values and the class value Output: the vector W of estimations of the qualities of attributes 1.Set all weights W[A]:-0:0; 2.For I := 1 to m do being 3.Randomly select an instance Ri; 4.Find nearest hit H and nearest miss M; 5.For A:= 1 to a do 6.W[A]: = W[A] -diff(A, Ri, II)/m +diff(A, Ri, M)/m 7.End

According to two results that were obtained, two hypotheses were proposed. Firstly, if there was a difference between the attribute values and the M value, the estimate quality W(A) was improved. Second, if there was a difference between the attribute values and the H value, the estimate quality W(A) was reduced [34]. The diff function in the following expressions determines the difference in values between the attribute A of two examples, I_1_ and I_2_.(3)$$diff\left(A,I_{1},I_{2}\right)=\frac{\left|value\left(A,I_{1}\right)-value\left(A,I_{2}\right)\right|}{\max(A)-\min(A)}$$

The maximum and minimum values of A were determined in all sets of samples. This normalization ensured that the quality of weight estimation for both discrete and continuous characteristics was between 0 and 1. Also, in the W [A] update, dividing the output of the diff function by m ensured that all final weights in the [−1, +1] range were normalized [35]. The diff function was also used to calculate the distance between samples when finding the nearest neighbors. The total distance was simply the sum of the diff intervals between all features (i.e., Manhattan distance) [36]. Technically, the original Relief algorithm used Euclidean distance instead of Manhattan distance, meaning that the diff expressions reached the power of two when measuring the sample distance and weighting the properties [37]. The normalized features obtained after this process can be effective in distinguishing between influenza and Newcastle disease in the head and leg areas of birds. This normalization made it possible to compare different characteristics such as color and texture on the same scale.

Figure 5 shows how the weights of the properties (W[A]) are updated by the Relief algorithm. In this algorithm, each property can have one of the discrete values X, Y, or Z, and the final result is binary (0 or 1).

When the algorithm compared a target instance with the nearest miss and hit instances, if the attribute value differed from the miss, the weight of that attribute increased by 1/m, and if it differed from the hit, the weight decreased by the same amount [37].

This weighting mechanism helped to identify the most effective characteristics in the diagnosis and classification of influenza and Newcastle disease, as the characteristics that make the largest distinction between different classes received heavier weights.

### 2.8. Machine Learning Classification and Evaluation

As demonstrated in the Introduction, the support vector machine classifier has demonstrated potential for poultry-related disease classification and was thus selected in this study for classifying avian influzenza and Newcastle disease, given the above extracted features. The model was trained with the three datasets: original thermal images, background-removed thermal images, and thermal images with chickens’ heads and legs only. A series of evaluation metrics were used to comprehensively evaluate the classification results.

One of the metrics was a confusion matrix. This matrix is an N × N square matrix, in which number of layers was N. For this study, there were 2 classes for sick birds and 1 class for healthy birds, resulting in a total of 3 classes (N = 3).

This matrix showed the relationship between the predicted and actual classes, where TP stands for the number of true positives, TN stands for the number of true negatives, FP stands for the number of false positives, and FN stands for the number of false negatives [38]. Based on the TP, TN, FP, and FN, the accuracy, precision, sensitivity, specificity, and F-measure were computed to evaluate the model’s classification performance (Equations (4)–(8)). Higher values of these metrics indicate better disease classification performance.(4)$$\mathrm{Accuracy}=\frac{\left(\mathrm{TP}+\mathrm{TN}\right)}{\left(\mathrm{TP}+\mathrm{TN}+\mathrm{FP}+\mathrm{FN}\right)}$$(5)$$\mathrm{Precision}=\frac{\mathrm{TP}}{\mathrm{TP}+\mathrm{FP}}$$(6)$$\mathrm{Sensitivity}=\frac{\mathrm{TP}}{\mathrm{TP}+\mathrm{FN}}$$(7)$$\mathrm{Specificity}=\frac{\mathrm{TN}}{\mathrm{TN}+\mathrm{FP}}$$(8)$$F-\mathrm{Measure}=\frac{2\times (\mathrm{Precision}\times \mathrm{Sensitivity})}{\left(\mathrm{Precision}+\mathrm{Sensitivity}\right)}$$

### 2.9. Integration of the Proposed Model with IoT for Real-Time Disease Monitoring

Integrating the proposed disease diagnosis model with an internet of things (IoT)-based real-time monitoring system could significantly increase its practical application in poultry farming. Such a system would enable the continuous monitoring of poultry health using thermal cameras and AI-based analysis tools on poultry farms. Using devices connected to the IoT, real-time thermal imaging data can be immediately processed and analyzed, allowing the early detection of symptoms of disease. In the event of a diagnosis, the system can automatically send alerts to farm operators and even poultry farms, enabling timely interventions and reducing the risk of outbreaks. In addition, integrating this system with cloud computing could facilitate remote monitoring and automated decision-making. This would allow veterinarians and farm managers to track the progression of diseases, analyze temperature trends, and actively implement preventive measures. Future research should focus on developing and testing an IoT prototype to assess its feasibility and effectiveness in real-world poultry farming conditions.

## 3. Results

### 3.1. Detection Accuracy of Head and Leg

The YOLO-v8 model achieved over 99% accuracy in detecting the heads and legs in thermal images and was able to obtain the coordinates of the enclosed bounding boxes, which were further used to crop the heads and legs from the thermal images. The high detection accuracy could be the result of simple and consistent image variations in the current settings. In another study, Yu et al. (2024) [39] used a cyclic transitional neural network (CCMNN) to eliminate cage effects and the YOLO-v8 algorithm to detect the crown and eyes of chickens in cage systems. The detection accuracy was improved by over 11% after they removed the cages from the images. Their results suggested that YOLO-v8 is a powerful model and could carry out the detection and cropping required in this study.

### 3.2. Classification Performance of Disease at Various Times Slots

Figure 6 shows the confusion matrix and Table 2 shows the evaluation criteria for detecting and classifying the three groups: original images, background-removed images, and images with chickens’ heads and legs only. The results of the table were divided into eight classes based on the infection time and within a time interval of 8 h. The average accuracy for the three groups of data was 75.89%, 83.93%, and 92.48%, respectively, for detecting avian influenza across all time slots. The accuracy was 83.04%, 91.52%, and 94.20%, respectively, for detecting Newcastle disease across all time slots. These results clearly outlined that removing the background or unnecessary pixels in the thermal images can boost the disease detection performance, and the performance can be even further improved when the model only focuses on the chickens’ heads and legs. The results also suggested that the classification accuracy for Newcastle disease was at least 2% higher than the one for avian influenza. This could be due to the faster disease transmission speed of Newcastle disease, which can cause more severe physiological and biological changes that may have provided supportive features for the model prediction.

According to Table 2, the classification accuracy for influenza disease at the different time slots (with 8 h intervals) of disease transmission was 92.59%, 92.31%, 92.86%, 90.00%, 89.29%, 96.30%, 90.00%, and 96.43%, respectively. The classification accuracy for Newcastle disease was 96.30%, 96.43%, 92.86%, 93.10%, 96.15%, 89.66%, 93.10%, and 96.43%, respectively. The results clearly suggested that, in looking at more focused regions (e.g., the head and leg) of thermal images, the machine learning classifier can identify poultry diseases early (e.g., within 16 h after disease infection). Qu et al. (2022) [40] used thermal imaging and machine learning methods to separate patients with lung infections and healthy people, achieving a classification accuracy of 93%. Pacheco et al. )2020 (identified the stress of lactating cows using thermal imaging and machine learning methods [41]. The proposed method had 80–90% accuracy in separating different cows with different health statuses. Hernández-Sánchez et al. )2024( [5] conducted a systematic review and meta-analysis of the various applications of the thermal imaging of the poultry under thermal stress. In general, four body areas (head, body, face, and leg) were identified as common areas of interest for measuring the body surface temperature of laying hens, broilers, and turkeys with thermal imagery. Areas of the body without feathers were valuable for heat regulation, which could be impacted by the age, species, thermal treatment, and naked body area of the subject.

Sadeghi et al. (2023) [18] used the thermal characteristics of the entire chicken body to diagnose Newcastle and influenza disease. The support vector machine classifier showed 80.56% and 86.11% accuracy for the classification of the two diseases within the first 8 h of infection. In contrast, our approach, focusing on head and leg features in thermal images, increased the classification accuracy to 94.20% and 92.48%, respectively, for the prediction of the two diseases. The improvement relied on selecting critical features for model learning and inference.

### 3.3. Prediction Performance of the Influenza and Newcastle Diseases via a Single Model

The previous section focuses on training the support vector machine classifier separately for the classification of the two diseases. Table 3 shows the detection accuracy (%) of the influenza and Newcastle diseases at different time points after infection via a single model. The best accuracy obtained for the three datasets was 86.90%, 88.10%, and 92.86%, respectively, at the 56th hour of disease infection. The results highlighted the necessity of focusing on regions of interest for disease classification. In addition, as the time since disease infection progressed, a higher classification accuracy was obtained, indicating the occurrence of more disease symptoms or features that supported the classification [42,43]. However, the results showed that the single model is more time-consuming than the chicken head and foot separation model in Section 3.2.

Table 4 shows the confusion matrix and evaluation criteria for the classification of influenza and Newcastle disease at all time points via a single model. Comparing the algorithm’s performance in the three types of images, the separation of the head and leg from the body showed the greatest impact on improving the accuracy of detection. This is most likely due to the focus on certain areas of the bird’s body that make the symptoms/features of the disease more obvious. Further, removing the background from the images also effectively increased the algorithm’s ability to diagnose the disease, as the processing accuracy increased as the visual noise decreased and the bird became more focused as a result. Previous studies [44,45] also indicated the necessity of preprocessing in improving the classification performance of a model. It should be noted that the single model performance was poorer than that of the multiple model at the same time slot. Perhaps, specific models have to be trained for detecting specific diseases being trained to detect all diseases.

Sadeghi et al. (2015) [46] showed that the use of voice-based diagnostic methods for chickens, especially in the diagnosis of Clostridium perfringe bacteria, is not of optimal accuracy or efficiency. In this method, the accuracy of diagnosis in the early stages of the disease was only 66.6%, and acceptable results were only achieved eight days after the onset of the disease. However, the thermal imaging method, focusing on the head and leg areas of chickens, was able to detect influenza and Newcastle disease with an accuracy of more than 92% in the first 16 h after infection. Another notable point is that acoustic methods require controlled laboratory conditions and limited sample numbers, but thermal imaging methods can be implemented in the real-world conditions of poultry farms and can simultaneously identify several diseases with high accuracy.

In recent years, thermal image processing has received a large amount of attention as an innovative tool in the diagnosis of poultry diseases. This technique, which is based on the identification of changes in the body temperature of poultry, is especially applicable to diseases whose initial symptoms are significantly associated with fever and changes in body temperature.

Yaji et al. (2023) [47] systematically reviewed the latest literature on the diagnosis of poultry disease based on deep learning techniques and clarified the clinical manifestations associated with various diseases.

Ahmed et al. (2021) [48] presented an artificial intelligence-based method for classifying healthy and sick chickens. The proposed method used the generative advertising networks method. The results show that the proposed method achieved 97% accuracy in separating sick and healthy chickens.

In another study [49], three versions of the YOLO-based algorithm (v8, v7, v5) were comparatively evaluated using thermal and enhanced visual datasets. The goal was to develop heat-based models for detecting broilers in complex environments and to classify pathological phenomena under challenging light conditions. After training on confirmed pathological phenomena, the YOLO-v8-based thermal model demonstrated exceptional performance and achieved the highest accuracy (mAP50 of 98.8% and F1 score of 97.2%) in the detection of objects among the tested methods. Verma et al. (2024) [50] used the EfficientNetB3 model to diagnose chicken diseases. Their approach used advanced preprocessing methods and strict training rules to train the model. All of the abovementioned studies, along with the current study, demonstrate the great potential of applying thermal images and deep learning techniques to poultry disease detection.

### 3.4. Feature Selection by the Relief Algorithm

Important features such as color and texture were extracted from the images with chickens’ heads and legs only. These features were rated by the Relief feature selection algorithm and are presented in Table 5. The 10 characteristics listed in Table 5 were selected as the most effective and key parameters for the diagnosis and classification of influenza and Newcastle disease and can be used as a practical resource for other researchers. It seemed that the share of texture-related features was more significant than that of other features. On the other hand, color information, especially in HSV space, can reflect text or color changes in related channels and was important for the classifications in this study. The choice of standard deviation in the hue channel of the HSV space (which is related to the color spectrum) was a testament to this importance [51].

Energy, as one of the important textural features in this study, is a measure of the pixel intensity variation in an image. This feature was calculated as the difference between a pixel’s intensity value and the average intensity of other pixels in the image. A higher energy value indicates a more distinct differentiation of a pixel from the rest of the image, and this distinction may indicate abnormal or disease-suspicious regions. Therefore, energy was an effective feature for classifying healthy and infected regions.

Entropy is another textural feature that indicates the level of complexity or disorder in an image. This feature is calculated as the negative sum of the pixel intensity value probabilities in an image [38,52]. Higher entropy values indicate that pixels in an image are more unpredictable and disordered, which was typically observed in disease-related regions. Thus, entropy acted as a key metric in distinguishing between natural and damaged regions and played a vital role in improving the classification accuracy [53].

Standard deviation is another important feature that measures the dispersion of pixel values. This feature is calculated as the root mean square difference between each pixel intensity value and the overall mean of the image’s intensity [52]. A higher standard deviation indicates the presence of more severe spread or variations in pixel intensity values. These changes typically occurred in regions that underwent thermal or color changes due to inflammation or infection. Therefore, this feature can be highly effective in detecting abnormal regions.

One of the key points in the analysis of thermal and color images was changes in the color intensity and temperature in contaminated areas. These changes occur due to inflammation, infection, or other physiological disorders in the chicken’s body. In particular, the changes in severity in the areas of the chickens’ heads and legs that were susceptible to disease were more pronounced than those in other areas of the body. This difference meant that the characteristics extracted from these areas, such as the standard deviation in the hue channel of the HSV color space and other features (e.g., energy and entropy), played a key role in the classification. In contrast, other areas of the body that lacked significant changes in thermal or color intensity were less important in diagnosing the diseases. Therefore, separating the chickens’ heads and legs from the entire image can have a significant impact on increasing the accuracy of diagnostic systems. Using an algorithm, such as deep learning algorithms, to isolate disease areas can significantly improve the classification accuracy.

This is especially important when the goal is to accurately identify nuances in the color intensity and temperature in thermal and color images. Previous research has shown that deep learning algorithms, such as YOLO, with a strong ability to extract features and differentiate normal and contaminated areas can play an important role in the development of disease detection systems. Finally, the characteristics selected in this study, such as the standard deviation, energy, and entropy, were identified as key tools for detecting influenza and Newcastle disease due to their high ability to differentiate between healthy and affected areas. These features did not only allow for the identification of areas suspected of disease, but also provide the basis for the development of more advanced algorithms and more accurate systems in this area.

### 3.5. Limitations and Future Works

In the field of early detection, the results of this study can be used to improve the mechanisms and algorithms used in the poultry monitoring and management systems of poultry farms in order to diagnose diseases early. Future work includes deploying the model in intelligent IoT systems to continuously monitor the health of herds. These systems allow 24 h monitoring of the behavior and physiological status of chickens using a network of sensors, thermal cameras and image processors. The data collected through this type of sensor network are transferred to a central system and analyzed using artificial intelligence algorithms to quickly identify issues and inform the poultry farmer of any abnormal changes in the behavior or body temperature of chickens that could be a sign of disease. There are several critical challenges that need to be solved before the full integration of this system. First, the model still requires a large number of images to generate a generalizable performance. The head and legs of a bird can be presented at different angles and can be partially visible/invisible due to overlapping and occlusion, which can significantly impact the model performance. Birds can raise their body temperature due to social interactions and other stressors that are not related to diseases, causing false positives. Multiple birds presenting in a single image could result in more challenging and diverse behaviors and interactions than a single bird in an image.

## 4. Conclusions

In this study, a new technique was presented to detect influenza and Newcastle disease in poultry using the YOLO-v8 algorithm, a support vector machine classifier, and thermal imaging. The results showed that focusing on the chicken’s head and legs significantly increased the accuracy of the diagnosis, with the accuracy of the influenza detection reaching 92.48% and that for Newcastle disease reaching 94.20%. The proposed framework can achieve over 90% disease classification accuracy within 8 h of disease infection. The removal of the background and thermal analysis of the head and leg areas played an important role in improving the performance of the algorithm due to rendering the temperature changes in these areas more obvious. In addition, characteristics such as the energy, entropy, and standard deviation in images, as criteria for measuring temperature intensity changes, had a significant impact on the identification of the diseases. Thermal changes of the heads and legs of the studied chickens made it possible to more accurately distinguish between diseases due to their direct association with the symptoms of the disease. In contrast, other parts of the body were less important in classification due to their relatively uniform temperature.

## Figures and Tables

**Figure 1 animals-15-01114-f001:**
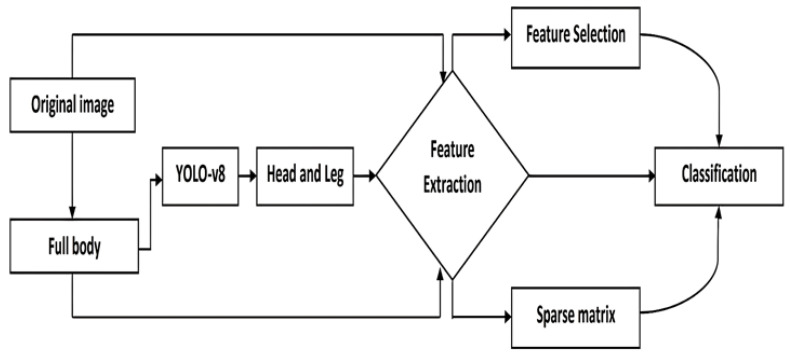
Research implementation steps.

**Figure 2 animals-15-01114-f002:**
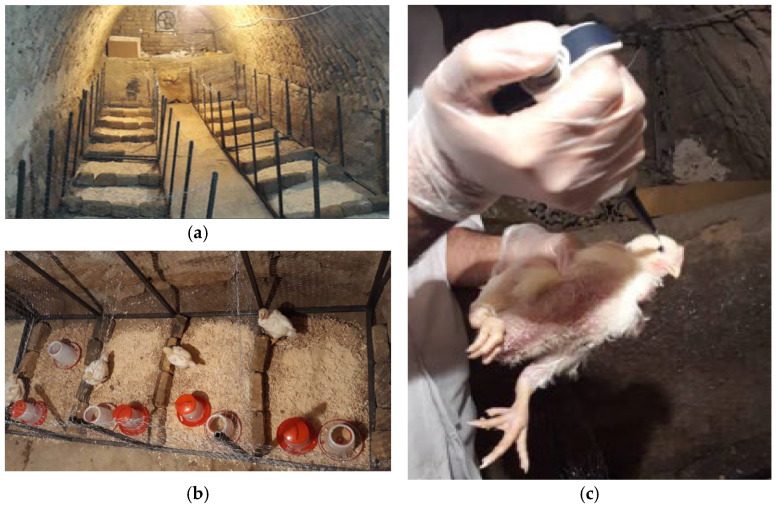
Photos of the experimental setup: (**a**) overall layout of the experimental pens; (**b**) detailed setup for each pen; and (**c**) eye drop for virus infection.

**Figure 3 animals-15-01114-f003:**
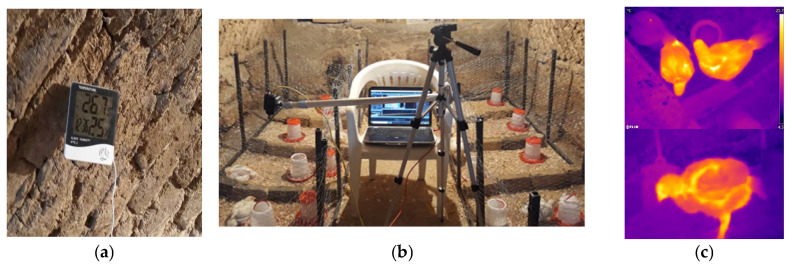
Illustration of data acquisition: (**a**) digital humidity and temperature meter; (**b**) setup of the thermal camera on the test site; and (**c**) two thermal image samples.

**Figure 4 animals-15-01114-f004:**
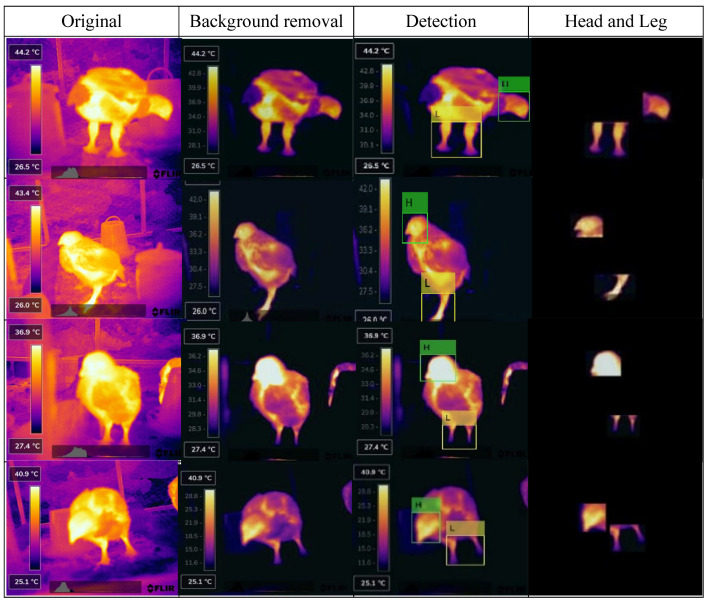
Preprocessed images of broiler chickens for detecting head and leg.

**Figure 5 animals-15-01114-f005:**
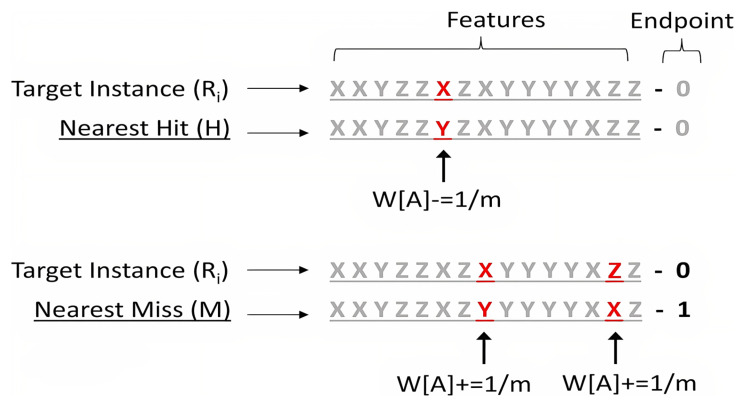
Updating the weight of features using the Relief algorithm.

**Figure 6 animals-15-01114-f006:**
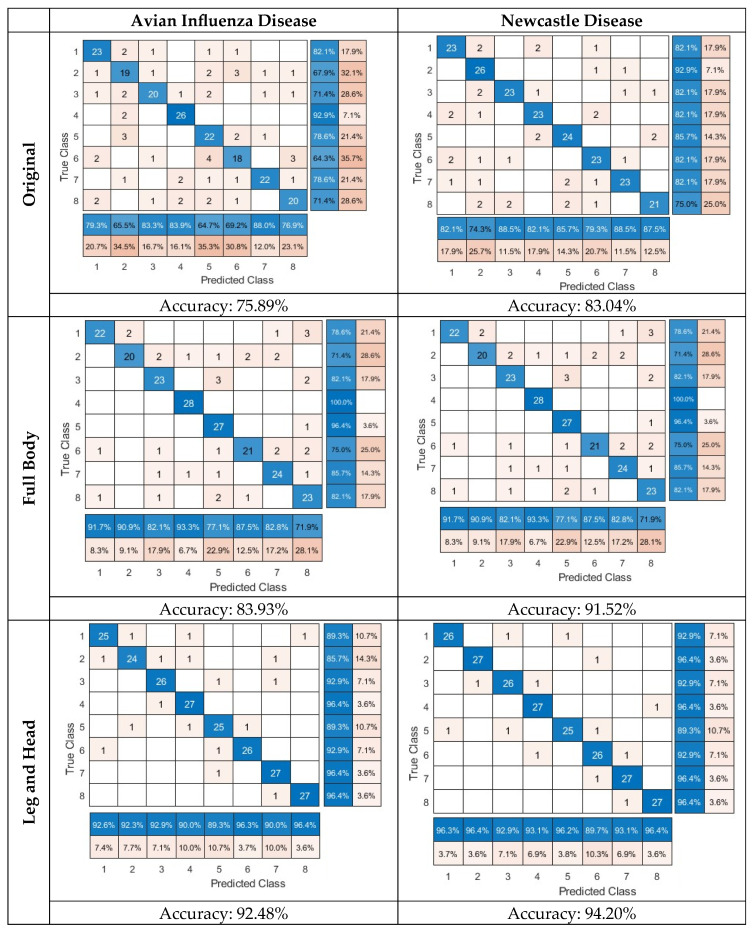
Confusion matrix of avian influenza and Newcastle disease at 8 (class 1), 16 (class 2), 24 (class 3), 32 (class 4), 40 (class 5), 48 (class 6), 56 (class 7) and 64 (class 8) hours.

**Table 1 animals-15-01114-t001:** Features and calculating formulas of extracted thermal images.

Feature	Description	Formulas	Reference
Mean	The average value can be calculated as the observed outcomes from the sample, divided by the total number of events.	$\;\bar{x}=\frac{1}{n}\sum_{i=1}^{n}x_{i}$	[22]
Max	The maximum lateral gray value can be assigned to the central pixel.	-	[24]
Standard Deviation	Obtaining a measure of how far data values are from the mean	$s=\sqrt{\frac{1}{n-1}\sum_{i=1}^{n}{(x-\;\bar{x}\;)}^{2}}$	[22]
Median	The middle value of a series of numbers is called the median, which represents the central value when the numbers are arranged in ascending or descending order	-	[24]
Contrast	The measurement can be expressed as the difference between the brightness of objects in a field of view	${\sum_{i,j}\left|i-j\right|}^{2}p(j,j)$	[23]
Correlation	The measure of the degree and type of relationship between adjacent pixels is referred to as spatial correlation	$\sum_{i,j}\frac{(i-\mu i)(j-\mu j)p(i,j)}{\sigma_{i}\sigma_{j}}$	[23]
Energy	The sum of the squared elements in the gray level co-occurrence matrix is known as energy, which represents the uniformity of the texture	$\sum_{i,j}{p(i,j)}^{2}$	[23]
Homogeneity	Element distribution in GLCM	$\sum_{i,j}\frac{p(i,j)}{1+\left|i-j\right|}$	[23]
Mean	Measuring the average pixel intensity numerically present in the region	$\frac{1}{n}\left(\sum_{i=1}^{n}X_{i}\right)$	[23]
Standard Deviation	Measuring the difference between gray levels and the average	$\sqrt{\frac{1}{n}\left(\sum_{i=1}^{n}{\left(X_{i}-\;\bar{X}\right)}^{2}\right)}$	[23]
Entropy	Measuring the difference in gray levels	E=sum(p.×log2(p))	[23]
Root Mean Square (RMS)	Measuring the root mean square value in an image	$X_{\mathrm{RMS}}=\sqrt{\frac{1}{N}{\sum_{n=1}^{N}\left|X_{n}\right|}^{2}}$	[23]
Variance	Measuring the numerical value of the variance of an image	$\frac{1}{n}\left({\sum_{i=1}^{n}\left(X_{i}-\;\bar{X}\right)}^{2}\right)$	[23]
Smoothness	Measuring the smoothness of the intensity ratio in an area	-	[23]
Kurtosis	Measuring the distribution of peaks associated with a normal distribution	$K=\frac{E(x-{\mu )}^{4}}{\sigma^{4}}$	[23]
Skewness	Measuring asymmetry in a statistical distribution	$S=\frac{E(x-{\mu )}^{3}}{\sigma^{3}}$	[23]

**Table 2 animals-15-01114-t002:** Performance evaluation parameters of disease prediction based on the thermal images with chickens’ heads and legs only.

Performance for Predicting the Influenza Disease								
	Healthy	Class 2	Class 3	Class 4	Class 5	Class 6	Class 7	Class 8
True Positives	25	24	26	27	25	26	27	27
False Positives	2	2	2	3	3	1	3	1
False Negatives	3	4	2	1	3	2	1	1
True Negatives	194	194	194	193	193	195	193	195
Precision (%)	92.59	92.31	92.86	90.00	89.29	96.30	90.00	96.43
Sensitivity (%)	89.29	85.71	92.86	96.43	89.29	92.86	96.43	96.43
Specificity (%)	98.89	98.98	98.98	98.47	98.47	99.49	98.47	99.49
Accuracy (%)	92.41	92.41	92.41	92.41	92.41	92.41	92.41	92.41
F-Measure (%)	90.91	88.89	92.86	93.10	89.29	94.55	93.10	96.43
**Performance for predicting the Newcastle disease**								
True Positives	26	27	26	27	25	26	27	27
False Positives	1	1	2	2	1	3	2	1
False Negatives	2	1	2	1	3	2	1	1
True Negatives	195	195	194	194	195	193	194	195
Precision (%)	96.30	96.43	92.86	93.10	96.15	89.66	93.10	96.43
Sensitivity (%)	92.86	96.43	92.86	96.43	89.29	92.86	96.43	96.43
Specificity (%)	99.49	99.49	98.98	98.98	99.49	98.47	98.98	99.49
Accuracy (%)	94.20	94.20	94.20	94.20	94.20	94.20	94.20	94.20
F-Measure (%)	94.55	96.43	92.86	94.74	92.59	91.23	94.74	96.43

**Notes:** class 2 = 16 h; class 3 = 24 h; class 4 = 32 h; class 5 = 40 h; class 6 = 48 h; class 7 = 56 h; and class 8 = 64 h.

**Table 3 animals-15-01114-t003:** Detection accuracy (%) of the influenza and Newcastle diseases at different hours after infections via a single model.

Time (h)	Original	Full Body	Leg and Head
8	42.86	45.24	48.81
16	46.43	53.57	54.76
24	51.19	53.57	60.71
32	55.95	57.14	64.29
40	64.29	65.48	72.62
48	67.86	70.24	80.95
56	86.90	88.10	92.86

**Table 4 animals-15-01114-t004:** Confusion matrix and evaluation criteria for bird disease prediction via a single model.

			Healthy	Avian Influenza	Newcastle Disease
Original	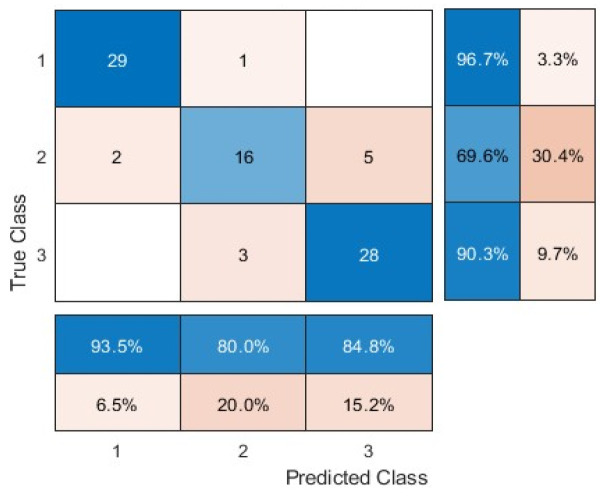	True Positives	29	16	28
		False Positives	2	4	5
		False Negatives	1	7	3
		True Negatives	52	57	48
		Precision (%)	93.55	76.67	84.85
		Sensitivity (%)	96.67	69.57	90.32
		Specificity (%)	96.30	93.44	90.57
		Accuracy (%)	86.90	86.90	86.90
		F-Measure (%)	95.08	74.42	87.50
	Accuracy: 86.90%				
			**Healthy**	**Avian influenza**	**Newcastle disease**
Full Body	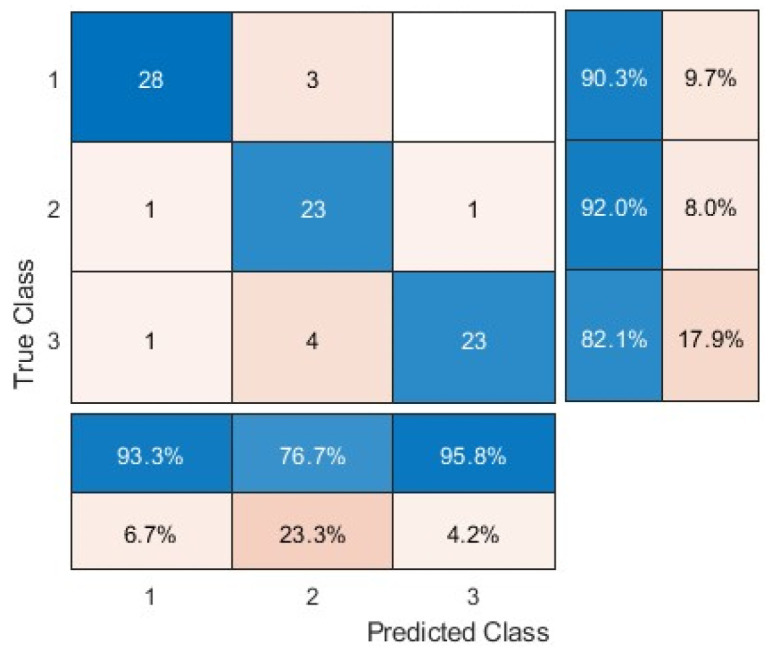	True Positives	28	23	23
		False Positives	2	7	1
		False Negatives	3	2	5
		True Negatives	51	52	55
		Precision (%)	93.33	80	95.83
		Sensitivity (%)	90.32	92	82.14
		Specificity (%)	96.23	88.14	98.21
		Accuracy (%)	88.10	88.10	88.10
		F-Measure (%)	91.80	83.64	88.46
	Accuracy: 88.10%				
Leg and Head	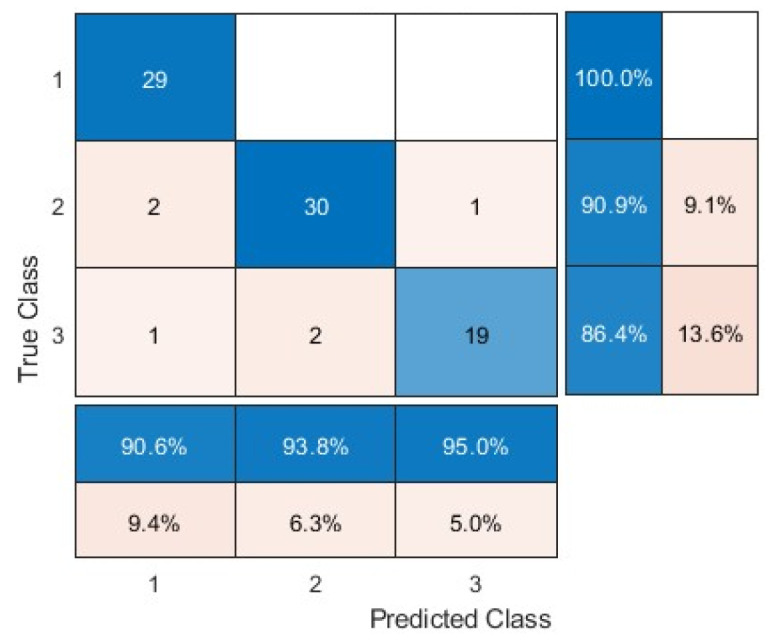		**Healthy**	**Avian influenza**	**Newcastle disease**
		True Positives	29	30	19
		False Positives	3	2	1
		False Negatives	0	3	3
		True Negatives	52	49	61
		Precision (%)	93.63	93.75	95
		Sensitivity (%)	100	90.91	86.36
		Specificity (%)	94.55	96.08	98.39
		Accuracy (%)	92.86	92.86	92.86
		F-Measure (%)	95.08	92.31	90.48
	Accuracy: 92.86%				

**Table 5 animals-15-01114-t005:** Feature selection results for the purpose of detecting and classifying influenza and Newcastle disease.

Features Rank	Features Type	Selected Feature	Selected Space (Band)
Feature 1	Texture	Energy	RGB (R)
Feature 2	Texture	Entropy	RGB (G)
Feature 3	Texture	Standard Deviation	HSV (H)
Feature 4	Color	Mean	LAB (L)
Feature 5	Color	Max	LAB (L)
Feature 6	Texture	Entropy	RGB (R)
Feature 7	Texture	Energy	HSV (S)
Feature 8	Color	Mean	RGB (R)
Feature 9	Texture	Standard Deviation	HSV (S)
Feature 10	Color	Max	LAB (A)

## Data Availability

The data will be available upon request.

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
