# Peer review of "Separating Chickens’ Heads and Legs in Thermal Images via Object Detection and Machine Learning Models to Predict Avian Influenza and Newcastle Disease"

_animals, 2025, doi:10.3390/ani15081114_

Round 1
Reviewer 1 Report
Comments and Suggestions for Authors
The article is devoted to the development of a non-invasive method for the early diagnosis of avian diseases (avian influenza and Newcastle disease) using thermal images and machine learning models. The authors apply the YOLO-v8 model to extract the head and legs of chickens from thermal images and the Support Vector Machine model to classify diseases. There are a few comments below.
1. Line 128: At the end of the section 'Introduction' it is necessary to add detailed information about the structure of the scientific article with references to the relevant sections.
2. Line 142: The quality of the image shown in Figure 1 should be improved. The caption of this figure should be more detailed and fully disclose the contents of the diagram.
3. Section 2.1. No information on the composition of the nutrient medium, if any, used. What solutions were used to keep the chicks in normal condition?
4. Line 169: The device model 'Sigma' is mentioned, but its characteristics (measurement accuracy, error limits) are missing. What is the accuracy of temperature and humidity measurements? It is necessary to specify this in the text of the article.
5. Line 172: What calibration mode was used in FLIR Tools? Was the camera pre-calibrated before the experiment?
6. Line 176: What parameters were used to assess the level of acclimatisation (behavioural change, feeding, physiological indicators, etc.)?
7. Line 178: Why was imaging performed every 8 hours? Is this interval based on previous studies, virus physiology, or technical limitations? This needs to be described in more detail in the article.
8. Line 179: How was the onset of visible symptoms of the disease determined? Were they standardised on a scale or subjectively assessed?
9. Line 191: How exactly were the histograms used? Was a specific segmentation algorithm applied?
10. Line 198: Was a pre-trained model used or was the training done from scratch?
11. Line 199: How many images were used for training and testing? What was the ratio of training and test samples?
12. Line 201: Need to specify which training parameters were used (number of epochs, batch size, loss function, etc.)?
13. Line 202: What dataset was the model tested on, what metric was used to measure accuracy (mAP, F1 score, Precision / recall)?
14. Line 208: The quality of the text in the captions in Figure 4 needs to be improved as the text looks muddy and stretched.
15. Line 263. Figure 5 should be deleted and its contents presented as a code listing.
16. Line 286. The quality of Figure 6 should be improved. It is not acceptable to publish images at this resolution.
17. Line 336. Is 2 percent a meaningful difference (e.g., confidence interval)?
Comments on the Quality of English LanguageThe text needs to be edited for correct sentence construction.
Author Response
Response to the Reviewers’ Comments:
The authors would like to thanking you for your valuable comments to improve the quality of the manuscript. The following points are corrected and highlighted in yellow in the manuscript.
Reviewer 1
The article is devoted to the development of a non-invasive method for the early diagnosis of avian diseases (avian influenza and Newcastle disease) using thermal images and machine learning models. The authors apply the YOLO-v8 model to extract the head and legs of chickens from thermal images and the Support Vector Machine model to classify diseases. There are a few comments below.
- Line 128: At the end of the section 'Introduction' it is necessary to add detailed information about the structure of the scientific article with references to the relevant sections.
Response. The text is corrected in the revised manuscript.
- Line 142: The quality of the image shown in Figure 1 should be improved. The caption of this figure should be more detailed and fully disclose the contents of the diagram.
(It is done)
- Section 2.1. No information on the composition of the nutrient medium, if any, used. What solutions were used to keep the chicks in normal condition?
(It is done)
- Line 169: The device model 'Sigma' is mentioned, but its characteristics (measurement accuracy, error limits) are missing. What is the accuracy of temperature and humidity measurements? It is necessary to specify this in the text of the article.
Response. The text is corrected in the revised manuscript.
- Line 172: What calibration mode was used in FLIR Tools? Was the camera pre-calibrated before the experiment?
Response. Thank you for your valuable feedback. The text is corrected in the revised manuscript.
- Line 176: What parameters were used to assess the level of acclimatisation (behavioural change, feeding, physiological indicators, etc.)?
Response. Thank you for your valuable question. The text is corrected in the revised manuscript.
- Line 178: Why was imaging performed every 8 hours? Is this interval based on previous studies, virus physiology, or technical limitations? This needs to be described in more detail in the article.
Response. Thank you for your valuable question. The text is corrected in the revised manuscript.
- Line 179: How was the onset of visible symptoms of the disease determined? Were they standardised on a scale or subjectively assessed?
Response. Thank you for your valuable question. The text is corrected in the revised manuscript.
- Line 191: How exactly were the histograms used? Was a specific segmentation algorithm applied?
Response. Thank you for your valuable question. The text is corrected in the revised manuscript.
- Line 198: Was a pre-trained model used or was the training done from scratch?
Response. Thank you for your valuable question. The text is corrected in the revised manuscript.
- Line 199: How many images were used for training and testing? What was the ratio of training and test samples?
Response. Thank you for your valuable question. The text is corrected in the revised manuscript.
- Line 201: Need to specify which training parameters were used (number of epochs, batch size, loss function, etc.)?
Response. Thank you for your valuable question. The text is corrected in the revised manuscript.
- Line 202: What dataset was the model tested on, what metric was used to measure accuracy (mAP, F1 score, Precision / recall)?
Response. Thank you for your valuable question. The text is corrected in the revised manuscript.
- Line 208: The quality of the text in the captions in Figure 4 needs to be improved as the text looks muddy and stretched.
(It is done)
- Line 263. Figure 5 should be deleted and its contents presented as a code listing.
(It is done)
- Line 286. The quality of Figure 6 should be improved. It is not acceptable to publish images at this resolution.
(It is done)
- Line 336. Is 2 percent a meaningful difference (e.g., confidence interval)?
Response. The text is revised
Reviewer 2 Report
Comments and Suggestions for Authors
The study provides a comprehensive approach to detecting avian influenza and Newcastle disease in poultry using thermal imaging and machine learning. This non-invasive method, focusing on the chicken's head and leg areas, is well-defined and highly relevant for early disease detection in the poultry industry. The use of YOLOv8 for object detection and the Support Vector Machine for classification is a promising combination that appears to yield strong results.
- Incorporating images from a wider variety of poultry species and breeds, as well as images from different environmental conditions, would help improve the model's adaptability and generalization. Additionally, data augmentation techniques could be employed to increase the variety of thermal images and reduce the model’s overfitting risk. (Section 2.1)
- A comparative analysis with other popular methods like conventional thermal imaging, or machine learning models like Convolutional Neural Networks (CNNs), could strengthen the evaluation of YOLOv8's performance. Such comparisons would provide a clearer understanding of the model's relative strengths and weaknesses in disease detection. (Section 4)
- Future work could focus on the integration of the disease detection model into a real-time monitoring system using Internet of Things (IoT) devices, which would allow continuous monitoring of poultry health and immediate disease detection alerts. (Section 4)
- The study focuses heavily on color and texture features extracted from the head and leg regions. Additional thermal features or techniques, such as deep learning-based feature extraction, could be explored to further improve the model’s accuracy and reduce computational costs. (Section 3.4)
- Since poultry farms often house multiple birds in one frame, future studies should focus on improving the model's performance when detecting multiple birds in complex images. Techniques such as instance segmentation or multi-object detection could be useful in addressing this challenge. (Section 2.8)
Author Response
Response to the Reviewers’ Comments:
The authors would like to thanking you for your valuable comments to improve the quality of the manuscript. The following points are corrected and highlighted in yellow in the manuscript.
Reviewer 2
Comments and Suggestions for Authors
The study provides a comprehensive approach to detecting avian influenza and Newcastle disease in poultry using thermal imaging and machine learning. This non-invasive method, focusing on the chicken's head and leg areas, is well-defined and highly relevant for early disease detection in the poultry industry. The use of YOLOv8 for object detection and the Support Vector Machine for classification is a promising combination that appears to yield strong results.
1) Incorporating images from a wider variety of poultry species and breeds, as well as images from different environmental conditions, would help improve the model's adaptability and generalization. Additionally, data augmentation techniques could be employed to increase the variety of thermal images and reduce the model’s overfitting risk. (Section 2.1)
Response. Thank you for your valuable feedback. We agree that the inclusion of images of different breeds of poultry, as well as imaging in varied environmental conditions, can help improve the generalization of the model. However, in this study, we focused on Ross 308 broiler chickens under controlled laboratory conditions to maintain stability in data collection and reduce the impact of environmental variables. This controlled setup ensured that the model learned disease-related thermal patterns without interference from external factors. Additionally, enhanced techniques were employed to improve model performance and mitigate overfitting risks. These methods increased the diversity of thermal imaging and contributed to the model’s generalization capability. Instead of relying on common data augmentation techniques, we introduced a novel approach by isolating the head and leg regions, which significantly improved model accuracy. This method enabled the model to focus on the most informative areas, eliminating unnecessary variations caused by full-body images. As a result, traditional data augmentation techniques, which offer limited novelty at this stage, were set aside, and a new proposed method was introduced to enhance model accuracy. The segmentation of the head and leg played a crucial role in boosting model performance, leading to higher precision and reliability in disease detection.
2) A comparative analysis with other popular methods like conventional thermal imaging, or machine learning models like Convolutional Neural Networks (CNNs), could strengthen the evaluation of YOLOv8's performance. Such comparisons would provide a clearer understanding of the model's relative strengths and weaknesses in disease detection. (Section 4)
Response. Thank you for your valuable suggestion. It's true that comparisons to traditional thermal imaging methods and machine learning models such as convolutional neural networks (CNNs) can enhance YOLO-v8 performance assessments. However, the main focus of this study was using YOLO-v8 to identify key areas (head and leg) for diagnosis, rather than making direct comparisons between different classification models. It is worth noting that previous studies have often used CNN-based methods to analyze the entire image of the chicken body. While our findings suggest that focusing on specific areas (head and leg) significantly improved diagnostic accuracy, this approach may not be directly comparable to whole-body-based methods.In future research, a comparative analysis with conventional thermal imaging and CNN-based models will be conducted to further assess the advantages and limitations of the proposed method. However, the primary objective of this study was to explore the impact of isolating the head and leg on disease detection accuracy, rather than evaluating different classification models.
3) Future work could focus on the integration of the disease detection model into a real-time monitoring system using Internet of Things (IoT) devices, which would allow continuous monitoring of poultry health and immediate disease detection alerts. (Section 4)
Response. Thank you for your kind offer. The text is modified in the revised manuscript. Lines 397 to 410.
4) The study focuses heavily on color and texture features extracted from the head and leg regions. Additional thermal features or techniques, such as deep learning-based feature extraction, could be explored to further improve the model’s accuracy and reduce computational costs. (Section 3.4)
Response. Absolutely true. In this study we used YOLO-v8 to extract the head and leg areas and we will continue to classification performance with deep learning-based feature engineering in the future.
5) Since poultry farms often house multiple birds in one frame, future studies should focus on improving the model's performance when detecting multiple birds in complex images. Techniques such as instance segmentation or multi-object detection could be useful in addressing this challenge. (Section 2.8)
Response. Thank you for your kind offer. This will definitely be considered in future research. This suggestion will be very helpful, especially in real-time detection.
Reviewer 3 Report
Comments and Suggestions for Authors
This paper studies an interesting problem of animal body detection and disease classification. The proposed approach separates chicken’s head and leg in thermal images via an object detection model and uses a machine learning model to predict avian influenza and newcastle disease.
There are two components in the proposed approach, including object detection and a classification. Currently, they are two separate models. Why not consider a multiple-task learning framework? That means the learned features from the YOLO object detector would be shared features between object detection and classification.
Experimental results could be enhanced in two aspects. The object detection performance evaluation is missing in the current manuscript. Secondly, it would be good to discuss how the feature selection (as shown in Table 5) could affect or adjust the proposed classification model.
In addition, Table 1 could be adjusted to reduce the column width so that its row heights can be reduced.
Author Response
Response to the Reviewers’ Comments:
The authors would like to thanking you for your valuable comments to improve the quality of the manuscript. The following points are corrected and highlighted in yellow in the manuscript.
Reviewer 3
Comments and Suggestions for Authors
This paper studies an interesting problem of animal body detection and disease classification. The proposed approach separates chicken’s head and leg in thermal images via an object detection model and uses a machine learning model to predict avian influenza and newcastle disease.
1)There are two components in the proposed approach, including object detection and a classification. Currently, they are two separate models. Why not consider a multiple-task learning framework? That means the learned features from the YOLO object detector would be shared features between object detection and classification.
Response. Yes, that's right. The text is modified in the revised manuscript (Lines 268 to 272).
2)Experimental results could be enhanced in two aspects. The object detection performance evaluation is missing in the current manuscript. Secondly, it would be good to discuss how the feature selection (as shown in Table 5) could affect or adjust the proposed classification model.
Response. Thank you for your valuable feedback. We have separately evaluated the detection accuracy for head and leg regions, ensuring a precise assessment of the YOLO-v8 model’s segmentation performance (Lines 416 to 419). The detection results for each region have been clearly presented in the manuscript. Regarding feature selection analysis, we have introduced the best and most significant features that contribute to disease classification. However, in this study, we have only focused on presenting these features and discussing their role in disease detection rather than analyzing the impact of each feature individually on classification accuracy. A more detailed investigation into the contribution of each selected feature will be considered in future studies (Lines 537 to 588).
3)In addition, Table 1 could be adjusted to reduce the column width so that its row heights can be reduced.
(It is done)
Reviewer 4 Report
Comments and Suggestions for Authors
This study proposes a method for detecting avian influenza and Newcastle disease in chickens using thermal imaging and machine learning. The research employs the YOLOv8 object detection model to isolate the head and leg regions in thermal images and then classifies disease presence using a Support Vector Machine (SVM) classifier. The study evaluates disease detection accuracy across three image types: original images, background-removed images, and images with only the head and leg regions. The results indicate that focusing on the head and leg significantly improves classification accuracy, reaching over 90% accuracy within 8 hours of infection. However, there are some major concerns that must be addressed before this work can be deemed suitable for publication.
1. Feature Selection Bias
• The feature selection process focused heavily on texture-based metrics, but other potentially useful features (e.g., temporal temperature variations) were not explored.
• The use of the Relief algorithm for feature selection is not justified against alternative techniques like PCA or deep-learning-based feature extraction.
2. Single Classification Model
• While SVM performs well in this study, deep learning classifiers such as CNNs might have been more effective in capturing complex thermal patterns and reducing the need for handcrafted features.
3. Lack of External Validation
• The study does not include external validation on an independent dataset, raising concerns about potential overfitting and real-world deployment performance.
4. Comparison to State-of-the-Art
• The study compares its results to prior works but lacks a clear benchmark against deep learning classifiers, which could offer a more direct comparison of model effectiveness.
5. Disease Progression Analysis
• While different infection time slots were considered, a more detailed temporal analysis (e.g., tracking individual birds over time) could provide insights into disease progression patterns.
6. Detection of Multiple Birds
• The study assumes single-bird images, but in real-world scenarios, multiple birds may be present in the same frame, requiring multi-object tracking or segmentation.
7. Thermal Camera Assumptions
• The study assumes a fixed emissivity setting (0.95), but variations in environmental conditions and individual bird characteristics could impact temperature readings.
8. Presentation and Formatting Issues
• Some sections contain repetitive explanations (e.g., importance of head and leg areas).
• Figures could be more clearly labeled and referenced for easier interpretation.
Author Response
Response to the Reviewers’ Comments:
The authors would like to thanking you for your valuable comments to improve the quality of the manuscript. The following points are corrected and highlighted in yellow in the manuscript.
Reviewer 4
Comments and Suggestions for Authors
This study proposes a method for detecting avian influenza and Newcastle disease in chickens using thermal imaging and machine learning. The research employs the YOLOv8 object detection model to isolate the head and leg regions in thermal images and then classifies disease presence using a Support Vector Machine (SVM) classifier. The study evaluates disease detection accuracy across three image types: original images, background-removed images, and images with only the head and leg regions. The results indicate that focusing on the head and leg significantly improves classification accuracy, reaching over 90% accuracy within 8 hours of infection. However, there are some major concerns that must be addressed before this work can be deemed suitable for publication.
- Feature Selection Bias
- The feature selection process focused heavily on texture-based metrics, but other potentially useful features (e.g., temporal temperature variations) were not explored.
Response. Thank you for your valuable feedback. The study focused on features that remain stable regardless of image scaling and spacing (fixed scale features) to ensure robustness in disease classification. The study did not take into account temporal temperature changes, as the goal was to identify consistent and reliable characteristics for detection. In addition, colored features were also extracted, but analysis showed that texture-based features had a significant impact on classification performance. Future studies could explore the integration of temporal features to further enhance the accuracy of the model.
- The use of the Relief algorithm for feature selection is not justified against alternative techniques like PCA or deep-learning-based feature extraction.
Response.That's right, and in this manuscript, the focus has been on the impact of head and leg parameters to diagnose and classify the disease. In future work, it must be done.
- Single Classification Model
Response. In this study, we used a linear Support Vector Machine (SVM) for classification. The choice of a linear model was based on its efficiency, interpretability, and lower computational cost compared to non-linear SVM variants.
One of the key advantages of a linear SVM is its ability to perform well when data is linearly separable, which simplifies the classification process and reduces the risk of overfitting, especially in high-dimensional feature spaces. Additionally, linear SVMs require fewer computational resources and are generally faster to train and deploy, making them more suitable for real-time or large-scale applications.
Although non-linear SVMs, such as those using Radial Basis Function or polynomial kernels, can handle more complex decision boundaries, they also increase computational complexity and may require extensive hyperparameter tuning. Given that texture-based features showed strong discriminative power in our study, a linear SVM was sufficient for achieving high classification accuracy while maintaining efficiency.
- While SVM performs well in this study, deep learning classifiers such as CNNs might have been more effective in capturing complex thermal patterns and reducing the need for handcrafted features.
Response. Thank you for your valuable feedback. You're absolutely right. While SVM performed well in this study, deep learning classifiers such as CNNs can be more effective at detecting complex thermal patterns and reduce the need for manual feature extraction. In fact, we're going to implement this system online in future models and take advantage of deep learning architectures.
In this study, our main focus was on the effective separation of the head and leg areas and to assess the impact of this approach on diagnostic accuracy. We will certainly use more advanced methods in the future, and we will publish the results of this research in the form of several manuscript that will make a comprehensive comparison between the different classification approaches in this area.
- Lack of External Validation
- The study does not include external validation on an independent dataset, raising concerns about potential overfitting and real-world deployment performance.
Response. You have made an important point. Yes, this study does not include external validation on an independent dataset, but we used five-fold cross-validation to reduce the overfitting risk and evaluate the generalization of the model.
This method allowed us to ensure that our models work well with different data and that our results are not random. By dividing the data into five separate sections and training and testing the model in five different iterations we were able to assess the stability of performance under different conditions.
- Comparison to State-of-the-Art
- The study compares its results to prior works but lacks a clear benchmark against deep learning classifiers, which could offer a more direct comparison of model effectiveness.
Response. Our results were compared with previous research to demonstrate the improvements in disease diagnosis using thermal imaging. However, we agree that direct comparisons with deep learning-based classification models, such as CNNs, could provide a more comprehensive evaluation of the proposed method’s efficiency.
Additionally, one of the main challenges of deep learning-based disease detection models is the requirement for large datasets. Several studies have demonstrated the need for extensive labeled data to achieve high classification accuracy. For instance, Ferentinos (2018) developed a deep learning model for plant disease detection using 87,848 images of 25 plant species in 58 different disease categories, collected under both laboratory and real field conditions. Similarly, Saleem et al. (2019) reviewed multiple deep learning architectures for plant disease classification and highlighted the importance of large-scale datasets such as PlantVillage, which contains 54,306 images of 14 different crops with 26 plant diseases. These studies emphasize that deep learning models often require large and diverse datasets to generalize effectively, which can be a significant limitation in certain applications.
Our primary focus was to develop a feature selection and classification-based approach, along with YOLO-v8 for precise segmentation of the head and leg regions. While CNN-based classifiers can automatically learn discriminative features, they require extensive labeled datasets and significant computational resources, which may pose limitations in real-world applications.
Given the limited availability of large-scale labeled thermal datasets, we focused on region-based analysis by isolating the most informative areas (head and legs) to improve disease detection accuracy. Future studies will explore deep learning-based classification methods and their integration with segmentation techniques to further enhance diagnostic performance.
- Disease Progression Analysis
- While different infection time slots were considered, a more detailed temporal analysis (e.g., tracking individual birds over time) could provide insights into disease progression patterns.
Response. Thank you for your valuable feedback. We agree that a detailed temporal analysis could provide valuable insights into disease progression patterns. However, unlike other chronic diseases, avian influenza is highly contagious. To prevent cross-contamination, animals infected with avian influenza are euthanized immediately, in accordance with animal use protocols. Given these constraints, conducting a continuous temporal analysis for avian influenza is not feasible.
- Detection of Multiple Birds
- The study assumes single-bird images, but in real-world scenarios, multiple birds may be present in the same frame, requiring multi-object tracking or segmentation.
Response. Yes, that's right. I confirm that in industrial farms, there are usually several chickens in an image frame that can pose a challenge in the detection of objects and the identification of diseases. In this study, our focus has been on single-bird analysis under controlled conditions to assess the effectiveness of the proposed method. However, we also understand the importance of expanding this method for more complex environments with the presence of multiple chickens in one future research, multi-object identification methods can be used to make a more accurate diagnosis.
- Thermal Camera Assumptions
- The study assumes a fixed emissivity setting (0.95), but variations in environmental conditions and individual bird characteristics could impact temperature readings.
Response. In this study, Emissivity was considered to be 0.95, which is a standard value for biological tissues, including poultry skin. The selection is based on previous research and instructions from the manufacturers of thermal cameras in animal studies.
- Presentation and Formatting Issues
- Some sections contain repetitive explanations (e.g., importance of head and leg areas).
Response. Various studies have shown that thermal imaging is an effective and non-invasive tool for monitoring poultry health and diagnosing diseases. Head and leg areas are known as key body temperature indicators in poultry. These areas are ideal areas for detecting temperature changes associated with diseases due to their high thermal exchange and the direct effect of diseases on blood circulation. In diseases such as the influenza and Newcastle, abnormal temperature changes are observed in these areas, which can help detect infection early and prevent its spread in breeding herds(Lines 124 to 138).
- Figures could be more clearly labeled and referenced for easier interpretation.
(It is done)
Round 2
Reviewer 1 Report
Comments and Suggestions for Authors
The revised manuscript can be accepted.
Reviewer 2 Report
Comments and Suggestions for Authors
Most of my comments are addressed. I recommend acceptance of this article.
Reviewer 3 Report
Comments and Suggestions for Authors
The revision is fine.
Reviewer 4 Report
Comments and Suggestions for Authors
Thank you for the revision and response. The reviewer’s concerns have been addressed.